# CLOSED-FORM SAMPLE PROBING FOR LEARNING GENERATIVE MODELS IN ZERO-SHOT LEARNING

**Samet Cetin, Orhun Bugra Baran & Ramazan Gokberk Cinbis**
Middle East Technical University
Department of Computer Engineering
Ankara, Turkey
`{cetin.samet,bugra.baran,gcinbis}@metu.edu.tr`

## ABSTRACT

Generative model based approaches have led to significant advances in zero-shot learning (ZSL) over the past few years. These approaches typically aim to learn a conditional generator that synthesizes training samples of classes conditioned on class definitions. The final zero-shot learning model is then obtained by training a supervised classification model over the real and/or synthesized training samples of seen and unseen classes, combined. Therefore, naturally, the generative model needs to produce not only relevant samples, but also those that are sufficiently *rich* for classifier training purposes, which is handled by various heuristics in existing works. In this paper, we introduce a principled approach for training generative models *directly* for training data generation purposes. Our main observation is that the use of closed-form models opens doors to end-to-end training thanks to the differentiability of the solvers. In our approach, at each generative model update step, we fit a task-specific closed-form ZSL model from generated samples, and measure its loss on novel samples all within the compute graph, a procedure that we refer to as *sample probing*. In this manner, the generator receives feedback directly based on the value of its samples for model training purposes. Our experimental results show that the proposed sample probing approach improves the ZSL results even when integrated into state-of-the-art generative models.

## 1 INTRODUCTION

*Zero-shot Learning* (ZSL) has recently received great interest for being one of the promising paradigms towards building very large vocabulary (visual) understanding models with limited training data. The problem of ZSL can be summarized as the task of transferring information across classes such that the instances of *unseen* classes, with no training examples, can be recognized at test time, based on the training samples of *seen* classes. Similarly, the term *Generalized Zero-Shot Learning* (GZSL) (Chao et al., 2016; Xian et al., 2018a) is used to refer to a practically more valuable variant of ZSL where both seen and unseen classes may appear at test time. GZSL brings in additional challenges since GZSL models need to produce confidence scores that are comparable across all classes.

Recent work shows that hallucinating unseen class samples through statistical generative models can be an effective strategy, *e.g.* (Mishra et al., 2017; Xian et al., 2018b; 2019; Sariyildiz & Cinbis, 2019; Zhu et al., 2018; Schonfeld et al., 2019; Arora et al., 2018; Narayan et al., 2020). These approaches rely on generative models conditioned on *class embeddings*, obtained from auxiliary semantic knowledge, such as visual attributes (Xian et al., 2019), class name word embeddings (Schonfeld et al., 2019), or textual descriptions (Zhu et al., 2018). The resulting synthetic examples, typically in combination with existing real examples, are used for training a supervised classifier.

In generative GZSL approaches, the *quality* of class-conditional samples is crucial for building accurate recognition models. It is not straight-forward to formally define the criteria of *good training samples*. Arguably, however, samples need to be (i) realistic (*e.g.* free from unwanted artifacts), (ii) relevant (*i.e.* belong to the desired class distribution) and (iii) informative (*i.e.* contain examples

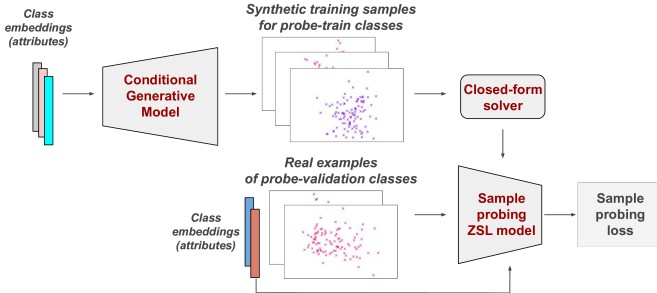

Figure 1: Illustration of the proposed framework for the end-to-end *sample probing* of conditional generative models. At each training iteration, we take synthetic training examples for some subset of seen classes (*probe-train classes*) from the conditional generative models, train a closed-form solvable zero-shot learning model (*sample probing ZSL model*) over them and evaluate it on the real examples of a different subset of seen classes (*probe-validation classes*). The resulting cross-entropy loss of the probing model is used as a loss term for the generative model update.

defining class boundaries) to train an accurate classifier. Clearly, a primary factor affecting the quality of generated samples is the loss driving the conditional generative model training process.

In this work, we aim to address the problem of training data generating models via an end-to-end mechanism that we call *sample probing*.[1] Our main goal is to *directly* evaluate the ability of a generative model in synthesizing training examples. To this end, we observe that we can leverage classification models with closed-form solvers to efficiently measure the quality of training samples, in an end-to-end manner. More specifically, we formulate a simple yet powerful meta-learning approach: at each training iteration, (i) take a set of samples from the generative model for a randomly selected subset of classes, (ii) train a zero-shot *probing model* using only the synthesized samples, and (iii) evaluate the probing model on real samples from the training set. We then use the loss value as an end-to-end training signal for updating the generative model parameters. Since we specifically focus on probing models with exact closed-form solutions, the probing model optimization simplifies into a differentiable linear algebraic expression and takes part as a differentiable unit within the compute graph. A graphical summary of the proposed training scheme is given in Figure 1.

In the rest of the paper, we first provide an overview of related work, and explain our approach in detail. We then present a thorough experimental evaluation on GZSL benchmarks in Section 4, where the results show that sample probing yields improvements when introduced into state-of-the-art baselines. We conclude with final remarks in Section 5.

## 2 RELATED WORK

The generalized zero-shot learning problem has been introduced by Xian et al. (2017) and Chao et al. (2016). The extensive study in Xian et al. (2018a) has shown that the success of methods can greatly vary across zero-shot and generalized zero-shot learning problems. The additional challenge in the generalized case is the need for deciding whether an input test sample belongs to a seen or unseen class. Discriminative training of ZSL models, such as those based on bilinear compatibility functions (Weston et al., 2011; Frome et al., 2013), are likely to yield higher confidence scores for seen classes. To alleviate this problem, a few recent works have proposed ways to regularize discriminative models towards producing comparable confidence scores across all classes and avoid over-fitting, *e.g.* Jiang et al. (2019); Liu et al. (2018); Chou et al. (2021b).

Generative approaches to zero-shot learning naturally address the confidence score calibration problem. However, in general, generative modeling corresponds to a more sophisticated task than learning discriminant functions only (Vapnik & Vapnik, 1998). In this context, the problem is further complicated by the need of predicting zero-shot class distributions. To tackle this challenging task, a variety of techniques have been proposed (Mishra et al., 2017; Xian et al., 2018b; 2019; Sariyildiz

---

[1]Our use of the *sample probing* term is not closely related to the natural language model analysis technique known as *probing* (Belinkov et al., 2017; Peters et al., 2018; Hewitt & Liang, 2019).

& Cinbis, 2019; Zhu et al., 2018; Schonfeld et al., 2019; Arora et al., 2018; Elhoseiny & Elfeki, 2019; Narayan et al., 2020) by adapting generative models, such as VAEs (Kingma & Welling, 2014; Rezende et al., 2014) and GANs (Goodfellow et al., 2014). To enforce class conditioning, the state-of-the-art approaches use a mixture of heuristics: Xian et al. (2018b) uses the loss of a pre-trained classifier over the generated samples during training. Narayan et al. (2020) additionally uses the loss of a sample-to-attribute regressor, in combination with a feedback mechanism motivated from *feedback networks* (Zamir et al., 2017). Sariyildiz & Cinbis (2019) uses *projection discriminator* (Miyato & Koyama, 2018) and *gradient matching loss* as a gradient based similarity measure for comparing real versus synthetic data distributions. None of these generative ZSL approaches, however, directly measure the *value* of the generated samples *for training* classification models. The main difficulty lies in the need for back-propagating over long compute chains, which is both ineffi-cient and prone to gradient vanishing problems. To the best of our knowledge, we introduce the first end-to-end solution to this problem by the idea of using probing models with closed-form solvers to monitor the sample quality for model training purposes.

Our approach is effectively a *meta-learning* (Snell et al., 2017) scheme. Meta-learning is a promi-nent idea in few-shot learning, where the goal is learning to build predictive models from a limited number of samples. The main motivation is the idea that general-purpose classification models may behave suboptimally when only a few training samples are provided. A variety of meta-learning driven few-shot learning models have been proposed, such as meta-models that transform few sam-ples to classifiers (Gidaris & Komodakis, 2018; Snell et al., 2017), set-to-set transformations for improving sample representations (Ye et al., 2020; Bronskill et al., 2020), fast adaptation networks for few examples (Finn et al., 2017; Rusu et al., 2019; Nichol et al., 2018). In contrast to such mainstream *learning to classify* and *learning to adapt* approaches, we aim to address the problem of *learning to generate training examples* for GZSL.

There are only a few and recent studies that aim to tackle generative ZSL via meta-learning prin-ciples. Verma et al. (2020; 2021) embrace the learning-to-adapt framework of MAML (Finn et al., 2017), which originally aims to learn the optimal initialization for few-shot adaptation. The MAML steps are incorporated by iteratively applying a single-step update to the generative model using the generative model (VAE/GAN) loss terms, and then back-propagating over the re-computed gen-erative loss terms on new samples from a disjoint subset of classes using the single-step updated model. Our approach differs fundamentally, as we propose to use the discriminative guidance of ZSL models fully-trained directly from generated sample batches. In another work, Yu et al. (2020) proposes an *episodic training*-like approach with periodically altered training sets and losses during training, to learn non-stochastic mappings between the class-embeddings and class-centers. Chou et al. (2021a), similarly inspired by meta-learning and mixup regularization (Zhang et al., 2018), proposes to train a novel discriminative ZSL model over episodically defined virtual training classes obtained by linearly mixing classes. Neither of these approaches learn sample generating models, therefore, they have no direct relation to our work focusing on the problem of measuring the sample quality for ZSL model training purposes, with end-to-end discriminative guidance.

The use of recognition models with closed-form solvers has attracted prior interest in various con-texts. Notably, Romera-Paredes & Torr (2015) proposes the ESZSL model as a simple and effective ZSL formulation. We leverage the closed-form solvability of the ESZSL model as part of our ap-proach. Bertinetto et al. (2019) utilizes ridge-regression based task-specific few-shot learners within a discriminative meta-learning framework. In a similar fashion, Bhat et al. (2020); Liu et al. (2020) tackle the problem of *video object segmentation* (VOS) and use ridge regression based task-specific segmentation models within a meta-learning framework. None of these approaches aim to use recog-nition models with closed form solvers to form guidance for generative model training.

Another related research topic is generative few-shot learning (FSL), where the goal is learning to synthesize additional in-class samples from a few examples, *e.g.* Hariharan & Girshick (2016); Wang et al. (2018); Gao et al. (2018); Schwartz et al. (2018); Lazarou et al. (2021). Among these, Wang et al. (2018) is particularly related for following a similar motivation of learning to generate good training examples. This is realized by feeding generated samples to a meta-learning model to obtain a classifier, apply it to real query samples, and use its query loss to update the generative model. Apart from the main difference in the problem definition (GZSL vs FSL), our work differs mainly by fully-training a closed-form solvable ZSL model from scratch at each training step, instead of a few-shot meta-learners that are jointly trained progressively with the generative model.

Finally, we note that meta-learning based approaches have also been proposed for a variety of different problems, such as learning-to-optimize (Li & Malik, 2016; Chen et al., 2017) and long-tail classification (Liu et al., 2019; Ren et al., 2020).

# 3 METHOD

In this section, we first formally define the generalized zero-shot learning problem and then define a mathematical framework to summarize the core training dynamics of mainstream generative GZSL approaches. We then express our approach in the context of this mathematical framework.

## 3.1 PROBLEM DEFINITION

In zero-shot learning, the goal is to learn a classification model that can recognize the test instances of *unseen* classes $\mathcal{Y}_u$, which has no training examples, based on the model learned over the training examples provided for the disjoint set of *seen* classes $\mathcal{Y}_s$. We refer to the class-limited training set by $\mathcal{D}_{tr}$, which consists of sample and class label pairs $(x \in \mathcal{X}, y \in \mathcal{Y}_s)$. In our work, we focus on ZSL models where $\mathcal{X}$ is the space of image representations extracted using a pre-trained ConvNet. In generalized zero-shot learning, the goal is to build the classification model using the training data set $\mathcal{D}_{tr}$, such that the model can recognize both seen and unseen class samples at test time. For simplicity, we restrict our discussion to the GZSL problem setting below.

In order to enable the recognition of unseen class instances, it is necessary to have visually-relevant prior knowledge about classes so that classes can visually be related to each other. Such prior knowledge is delivered by the mapping $\psi : \mathcal{Y} \to \mathcal{A}$, where $\mathcal{A}$ expresses the prior knowledge space. In most cases, the prior knowledge is provided as $d_\psi$-dimensional vector-space embeddings of classes, obtained using visual attributes, taxonomies, class names combined with word embedding models, the textual descriptions of classes combined with language models, see *e.g.* Akata et al. (2015). Following the common terminology, we refer to $\psi$ as the *class embedding* function.

**Generative GZSL.** In our work, we focus on generative approaches to GZSL. The main goal is to learn a conditional generative model $G : \mathcal{A} \times \mathcal{Z} \to \mathcal{X}$, which takes some class embedding $a \in \mathcal{A}$ and stochasticity-inducing noise input $z$, and yields a synthetic sample $x \in \mathcal{X}$. Once such a generative model is learned, synthetic training examples for all classes can simply be sampled from the $G$-induced distribution $P_G$, and the final classifier over $\mathcal{Y}$ can be obtained using any standard supervised classification model. We refer to the trainable parameters of the model $G$ by $\theta_G$.

As summarized in Section 2, existing approaches vary greatly in terms of their generative model details. For the purposes of our presentation, most of the GZSL works (if not all) can be summarized as the iterative minimization of some loss function that acts on the outputs of the generative model:

$$L_G = \mathop{\mathbb{E}}_{(x,a) \sim \mathcal{D}_{tr}} \left[ \ell_G \left( G(a, z_x), a \right) \right] \tag{1}$$

where $z_x$ refers to the noise input associated with the training sample $(x, y)$ and $\ell_G$ is the generative model learning loss. $(x, a) \sim \mathcal{D}_{tr}$ is a shorthand notation for $(x, \psi(y)) \sim \mathcal{D}_{tr}$ At each iteration, the goal is to reduce $L_G$ approximated over a mini-batch of real samples and their class embeddings.

In our notation, we deliberately keep certain details simple. Noticeably, $z_x$ greatly varies across models. For example, in the case of a conditional GAN model, $z_x \sim p(z)$ can simply be a sample from a simple prior distribution $p(z)$, *e.g.* as in (Xian et al., 2018b; Sariyildiz & Cinbis, 2019; Elhoseiny & Elfeki, 2019). In contrast, in variational training, $z_x$ is the latent code sampled from a variational posterior, *i.e.* $z_x \sim q(z|x)$, where the variational posterior $q(z|x)$ is given by a variational encoder trained jointly with $G$, *e.g.* as in (Narayan et al., 2020; Xian et al., 2019).

Another important simplification that we intentionally make in Eq. (1) is the fact that we define the generative model learning loss $\ell_G$ as a function of generator output and class embedding, to emphasize its sample-realisticity and class-relevance estimation goals. However, the exact domain of $\ell_G$ heavily depends on its details, which typically consists of multiple terms and/or (adversarially) trained models. In most of the state-of-the-art approaches, this term is a combination of VAE reconstruction loss (Xian et al., 2019), conditional or unconditional adversarial discriminator network (Xian et al., 2018b; Narayan et al., 2020; Sariyildiz & Cinbis, 2019; Xian et al.,

2019), a sample-to-class classifier for measuring class relevance (Xian et al., 2018b) and sample-to-embedding mappings (Narayan et al., 2020). The loss $L_G$ may also incorporate additional regularization terms, such as $\ell_2$ regularization or a gradient penalty term (Sariyildiz & Cinbis, 2019).

In the following, we explain our sample probing approach as a loss term that can, in principle, be used in conjunction with virtually any of the mainstream generative zero-shot learning formulations.

## 3.2 SAMPLE PROBING AS GENERATIVE MODEL GUIDANCE

The problem that we aim to address is the enforcement of $G$ to learn to produce samples maximally beneficial for zero-shot model training purposes. We approach this problem through a *learning to generate training samples* perspective, where we aim to monitor the quality of the generative model through the synthetic class samples it provides.

In the construction of our approach, at each training iteration $t$, we first randomly select a subset of $Y_{\text{pb-tr}}^t \subset \mathcal{Y}$ of seen classes. We refer to these classes as *probe-train* classes. This subset defines the set of classes that are used for training the iteration-specific *probing model* over the synthetic samples. More specifically, we first take samples from the model $G$ with the parameters $\theta_G^t$ for these classes, and fully train a temporary ZSL model over them using regularized loss minimization:

$$\Gamma^t = \arg\min_\Gamma \mathop{\mathbb{E}}_{x=G(z,a \sim A_{\text{pb-tr}}^t)} [\ell_{\text{pb}}(f_{\text{pb}}(x,a),a)] \tag{2}$$

where $f_{\text{pb}}$ is the scoring function of the temporary probing model parameterized by $\Gamma$ and $\ell_{\text{pb}}$ is its training loss. $A_{\text{pb-tr}}^t$ is the set of class embeddings of classes in $Y_{\text{pb-tr}}^t$. Regularization term over $\Gamma$ is not shown explicitly for brevity.

The result of Eq. (2), gives us a purely synthetic sample driven model $\Gamma^t$, which we leverage as a way to estimate the success of the generator in synthesizing training examples. For this purpose, we sample real examples from the training set $\mathcal{D}_{\text{tr}}$ as validation examples for the probe model. Since we use a (G)ZSL model as the probing model, we can evaluate the model on examples of the classes not used for training the model. Therefore, we sample these probe-validation examples from the remaining classes $Y_{\text{pb-val}} = \mathcal{Y}_s \setminus Y_{\text{pb-tr}}^t$, *i.e.* the classes with real training examples but unused for probe model training, and use softmax cross-entropy loss over these samples as the probing loss:

$$L_{\text{pb}} = - \mathop{\mathbb{E}}_{(x,y) \sim \mathcal{D}_{\text{pb-val}}} [\log p(y|x; \Gamma^t)] \tag{3}$$

where $\mathcal{D}_{\text{pb-val}} \subset \mathcal{D}_{\text{tr}}$ is the data subset of classes $Y_{\text{pb-val}}$. $p(y|x; \Gamma^t)$ is the target class likelihood obtained by applying softmax to $f_{\text{pb}}(x, \psi(y); \Gamma^t)$ scores over the set of target class set. Here, as the target class set, one can use only the classes in $Y_{\text{pb-val}}$ (ZSL probing) or those in both $Y_{\text{pb-tr}}$ and $Y_{\text{pb-val}}$ (GZSL probing). We treat this decision as a hyper-parameter and tune on the validation set.

We use a weighted combination of $L_{\text{pb}}$ and $L_G$, as our final loss function. Therefore, the gradients $\nabla_{\theta_G} L_{\text{pb}}$ act effectively as the training signal for guiding $G$ towards yielding training examples that result in (G)ZSL probing models with minimal empirical loss.

## 3.3 CLOSED-FORM PROBE MODEL

A critical part of the construction is the need for a probing model where minimization of Eq. (2) is both efficient and differentiable, so that the solver itself can be a part of the compute graph. Probing models that require iterative gradient descent based optimization are unlikely to be suitable as one would need to make a large number of probing model updates for each single $G$ update step, which is both inefficient and prone to gradient vanishing problems. We address this problem through the use of a ZSL model that can be efficiently fit using a closed-form solution.

For this purpose, we opt to use the ESZSL (Romera-Paredes & Torr, 2015) as the main closed-form probe model in our experiments. The model is formalized by the following minimization problem:

$$\min_\Gamma \|X^\mathsf{T} \Gamma A - Y\|_{\text{Fro}}^2 + \Omega(\Gamma) \tag{4}$$

where $X \in R^{d_x \times m}$ and $A \in R^{d_\psi \times k}$ represent the feature and class embeddings corresponding to $m$ input training examples and $k$ classes, $Y$ is the $\{0,1\}^{m \times k}$ matrix of groundtruth labels, $\Gamma \in R^{d_x \times d_\psi}$

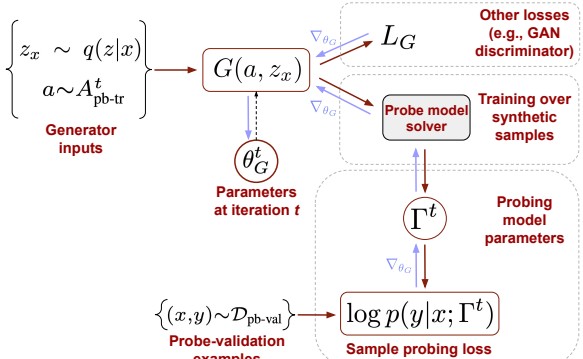

Figure 2: The compute graph view of the proposed approach, at some training iteration $t$. The upper half shows sampling from the generative model and the lower half shows sample probing model loss. Circles denote generator and probing model parameters. Blue arrows show the back-propagation path for updating the generative model. Best viewed in color.

is the compatibility model matrix, and $\Omega\left(\Gamma\right)$ is the regularization function, defined as:

$$\Omega\left(\Gamma\right) = \lambda_x\|\Gamma A\|_{\text{Fro}}^2 + \lambda_a\|X^{\text{T}}\Gamma\|_{\text{Fro}}^2 + \lambda_n\|\Gamma\|_{\text{Fro}}^2 \tag{5}$$

where $\lambda_x, \lambda_a, \lambda_n$ correspond to term weights. When the regularization term weights are set such that $\lambda_n = \lambda_x\lambda_a$, the optimal solution to Eq. (4) can be computed in a closed-form:

$$\Gamma^\star = (XX^{\text{T}} + \lambda_x I)^{-1}XYA^{\text{T}}(AA^{\text{T}} + \lambda_a I)^{-1} \tag{6}$$

This approach was originally proposed as a standalone label-embedding based ZSL model in Romera-Paredes & Torr (2015), with the practical advantage of having an efficient solver. Here, we re-purpose this approach as a probing model in our framework, where the fact that the model is solvable in closed-form is critically important, enabling the idea of end-to-end sample probing. For this purpose, we utilize the solver given by Eq. (6) as the implementation of Eq. (2), which takes a set of synthetic training samples and estimates the corresponding probing model parameters.

**Alternative probe models.** While we utilize ESZSL in our main experiments, we demonstrate the possibility of using the proposed approach with different probe models using two additional alternatives. The first one, which we call *Vis2Sem*, is the regression model from visual features to their corresponding class embeddings, defined as follows (using the same notation as in ESZSL):

$$\min_{\Gamma} \|\Gamma^{\text{T}}X - AY^{\text{T}}\|_{\text{Fro}}^2 + \lambda_n\|\Gamma\|_{\text{Fro}}^2. \tag{7}$$

A discussion of the Vis2Sem model can be found in Kodirov et al. (2017). The second one, which we call *Sem2Vis*, is the class embeddings to visual features regression model of Shigeto et al. (2015):

$$\min_{\Gamma} \|X - \Gamma AY^{\text{T}}\|_{\text{Fro}}^2 + \lambda_n\|\Gamma\|_{\text{Fro}}^2. \tag{8}$$

Both models, just like ESZSL, are originally defined as non-generative ZSL models, and we re-purpose them to define our data-dependent generative model training losses. Unlike the bi-linear compatibility model of ESZSL, however, these models rely on distance based classification, and do not directly yield class probability estimates. While one can still obtain a probability distribution over classes, *e.g.* by applying softmax to negative $\ell_2$ distances, for simplicity, we directly use the Sem2Vis and Vis2Sem based distance predictions between the visual features of probe-validation samples and their corresponding class embeddings to compute $L_{\text{pb}}$ as a replacement of Eq. (3).

**Summary.** A summary of the final approach from a compute graph point of view, is given in Figure 2. The proposed approach aims to realize the goal of learning to generate *good* training samples by evaluating the synthesis quality through the lens of a closed-form trainable probe model, the prediction loss of which is used as a loss for the $G$ updates. Therefore, $G$ is expected to be progressively guided *towards* producing realistic, relevant and informative samples, through the reinforcement of which may vary depending on the inherent nature of the chosen probe model.

# 4 EXPERIMENTS

**Datasets.** We use the four mainstream GZSL benchmark datasets: Caltech-UCSD-Birds (CUB) (Wah et al., 2011), SUN Attribute (SUN) (Patterson & Hays, 2012), Animals with Attributes 2 (AWA2, more simply AWA) (Xian et al., 2018a) and Oxford Flowers (FLO) (Nilsback & Zisserman, 2008). CUB is a medium scale fine-grained dataset consisting of 200 bird species classes, 11.758 images and 312 dimensional attribute annotations. SUN is a fine-grained dataset with 14.340 images and 102 dimensional attributes. AWA is a coarse-grained dataset with 30.475 images of 50 animal classes with 85-dimensional attributes. FLO is a medium-scale fine-grained dataset with 102 classes and 1024-dimensional attributes. Following the state-of-the-art, we use the class embeddings and the version-2 splits defined by Xian et al. (2018a). We evaluate the results in terms of GZSL-u, GZSL-s and h-score (H) values (Xian et al., 2018a). The h-score, *i.e.* the harmonic mean of GZSL-u and GZSL-s scores, aims to measure how well a model recognizes seen and unseen classes collectively.

In our experiments, we use the image features extracted from ResNet-101 backbone pretrained on ImageNet 1K. In the experiments based on *fine-tuned* representations, we use the backbone fine-tuned with the training images of seen classes, as in Xian et al. (2019); Narayan et al. (2020).

**Hyper-parameter tuning policy.** In our preliminary studies, we observe that the final GZSL performance, especially in terms of h-score, of most models strongly depends on the selection of the hyper-parameters. We also observe that there is no widely-accepted policy on how the hyper-parameters of GZSL models shall be tuned. It is a rather common practice in the GZSL literature to either directly report the hyper-parameters used in experiments without an explanation on the tuning strategy or simply refer to *tuning on the validation set*, which we find a vaguely-defined policy as (i) Xian et al. (2018a) defines an unseen-class only validation split, which does not allow monitoring the h-score, and, (ii) it is unclear which metric one should use for GZSL model selection purposes.

Therefore, to obtain comparable results within our experiments, we use the following policy to tune the hyper-parameters of our approach and our baselines: we first leave-out 20% of train class samples as *val-seen* samples. We periodically train a supervised classifier by taking synthetic samples from the generative model, and evaluate it on the validation set, consisting of the aforementioned *val-seen* samples plus the *val-unseen* samples with respect to the benchmark splits. We choose the hyper-parameter combination with the highest h-score on the validation set. We obtain final models by re-training the generative model from scratch on the training and validation examples combined.

## 4.1 MAIN RESULTS

In this subsection, we discuss our main experimental results. As we observe that the results are heavily influenced by hyper-parameter tuning strategy, our main goal throughout our experiments is the validation of the proposed sample probing idea by integrating it into strong generative GZSL baselines, and then comparing results using the same tuning methodology. Using this principle, we present two main types of analysis: (i) the evaluation of the proposed approach using ESZSL as the probe model in combination with a number of generative GZSL models, and (ii) the evaluation of alternative closed-form probe models within our framework.

**Generative GZSL models with sample probing.** To evaluate the sample probing approach as a general technique to improve generative model training, we integrate it to four recent generative GZSL approaches: conditional Wasserstein GAN (cWGAN) (Arjovsky et al., 2017; Miyato & Koyama, 2018), LisGAN (Li et al., 2019), TF-VAEGAN (Narayan et al., 2020) and FREE (Chen et al., 2021). We additionally report results for the variant of TF-VAEGAN with the fine-tuned representations (*TF-VAEGAN-FT*), as it is the only one among them with reported fine-tuning results. For the cWGAN, we follow the implementation details described in Sariyildiz & Cinbis (2019), and tune hyper-parameters using our policy. For LisGAN, TF-VAEGAN and FREE models, we use the official repositories shared by their respective authors. We use the version of TF-VAEGAN without *feedback loop* (Narayan et al., 2020), for simplicity, as the model yields excellent performance with and without feedback loop. In all models (except cWGAN), we only re-tune the number of training iterations of the original models using our hyper-parameter tuning policy, to make the re-

Table 1: **Evaluation of sample probing with multiple generative GZSL models on four benchmark datasets.** Each row pair shows the effect of adding sample probing to a particular generative GZSL model, using ESZSL as the closed-form probe model. We use the same hyper-parameter optimization policy in all cases to make results comparable. We observe h-score improvements at varying degrees in 17 out of 19 model, feature & dataset variations.

| | Sample probing | CUB | | | FLO | | | SUN | | | AWA | | |
|---|---|---|---|---|---|---|---|---|---|---|---|---|---|
| | | u | s | **H** | u | s | **H** | u | s | **H** | u | s | **H** |
| cWGAN (Miyato & Koyama, 2018) | N | 45.1 | 53.1 | 48.7 | 50.7 | 74.3 | 60.3 | 41.6 | 37.3 | 39.3 | - | - | - |
| | Y (ESZSL) | 48.2 | 52.4 | **50.2** | 51.8 | 74.1 | **61.0** | 44.4 | 36.6 | **40.1** | - | - | - |
| LisGAN (Li et al., 2019) | N | 40.9 | 60.5 | 48.8 | 53.1 | 81.7 | 64.4 | 41.5 | 36.6 | 38.9 | 44.2 | 77.0 | 56.1 |
| | Y (ESZSL) | 44.2 | 59.2 | **50.6** | 56.7 | 77.8 | **65.6** | 44.0 | 35.4 | **39.2** | 46.2 | 71.5 | **56.2** |
| TF-VAEGAN (Narayan et al., 2020) | N | 53.9 | 58.4 | 56.0 | 59.4 | 78.3 | 67.5 | 42.9 | 39.3 | 41.0 | 54.4 | 75.2 | 63.2 |
| | Y (ESZSL) | 51.1 | 63.3 | **56.6** | 63.5 | 83.2 | **72.1** | 44.0 | 39.7 | **41.7** | 55.2 | 74.7 | **63.5** |
| TF-VEAGAN-FT (Narayan et al., 2020) | N | 64.2 | 72.7 | 68.2 | 70.0 | 91.3 | 79.2 | 46.5 | 41.7 | **44.0** | 41.7 | 90.2 | 57.0 |
| | Y (ESZSL) | 63.1 | 76.1 | **69.0** | 70.2 | 91.7 | **79.5** | 47.8 | 40.6 | 43.9 | 45.6 | 87.6 | **60.0** |
| FREE (Chen et al., 2021) | N | 51.2 | 61.5 | **55.9** | 62.8 | 80.7 | 70.6 | 46.2 | 37.2 | 41.2 | 48.2 | 78.7 | 59.8 |
| | Y (ESZSL) | 51.6 | 60.4 | 55.7 | 65.6 | 82.2 | **72.9** | 48.2 | 36.5 | **41.5** | 51.3 | 78.0 | **61.8** |

Table 2: **Sample probing with alternative closed-form models**, based on TF-VAEGAN.

| Closed-form probe model | CUB | | | FLO | | | SUN | | | AWA | | |
|---|---|---|---|---|---|---|---|---|---|---|---|---|
| | u | s | **H** | **u** | s | **H** | **u** | s | **H** | **u** | s | **H** |
| - | 53.9 | 58.4 | 56.0 | 59.4 | 78.3 | 67.5 | 42.9 | 39.3 | 41.0 | 54.4 | 75.2 | 63.2 |
| ESZSL | 51.1 | 63.3 | 56.6 | 63.5 | 83.2 | **72.1** | 44.0 | 39.7 | 41.7 | 55.2 | 74.7 | **63.5** |
| Sem2Vis | 51.9 | 63.0 | **56.9** | 58.6 | 80.9 | 68.0 | 44.7 | 38.4 | 41.3 | 54.9 | 74.6 | 63.2 |
| Vis2Sem | 37.1 | 70.4 | 48.6 | 58.3 | 80.1 | 67.5 | 46.0 | 40.1 | **42.8** | 55.3 | 74.3 | 63.4 |

sults comparable, as it is unclear how the original values were obtained.[2] We keep all remaining hyper-parameters unchanged to remain as close as possible to the original implementations.

The results over the four benchmark datasets are presented in Table 1. In terms of the h-scores, we observe improvements in 17 out of 19 cases, at varying degrees (up to $4.6$ points ). Only in two cases we observe a slight degradation (maximum of $0.2$ points) in performance. Overall, these improvements over already strong and state-of-the-art (or competitive) baselines validate the effectiveness of the proposed sample probing approach, suggesting that it is a valid method towards end-to-end learning of generative GZSL models directly optimized for synthetic train data generation purposes.

**Sample probing with alternative closed-form models.** We now evaluate our approach with different closed-form probe models, specifically ESZSL, Sem2Vis (Shigeto et al., 2015) and Vis2Sem (Kodirov et al., 2017), as described in Section 3.3. For these experiments, we use the TF-VAEGAN as the base generative model.

The results with four configurations over four benchmark datasets are presented in Table 2. First of all, in terms of h-scores, we observe considerable performance variations across the probe models and datasets: Sem2Vis performs the best on CUB ($+0.9$ over the baseline), ESZSL provides a clear gain on FLO ($+4.6$) and a relative improvement on AWA ($+0.3$), and Vis2Sem improves the most on SUN ($+1.8$). These results suggest that sample probe alternatives have their own advantages and disadvantages, and their performances can be data dependent. Therefore, in a practical application, probe model options can be incorporated into the model selection process. More in-depth understanding of closed-form model characteristics for sample probing purposes, and the formulation and evaluation of other probe models can be important future work directions. Overall, the fact that we observe equivalent (2) or better (9) h-scores in 11 out of 12 sample probing experiments indicates the *versatility* of the approach in terms of compatibility with various closed-form probe models.

**Comparison to other generative GZSL approaches.** Performance comparisons across independent experiment results can be misleading due to differences in formulation-agnostic implementation and model selection details. Nevertheless, an overall comparison can be found in Appendix A.

---

[2]We also tune LisGAN for AWA2 as the original paper reports AWA1 results instead.

Table 3: ZSL vs GZSL based sample probing losses (using TF-VAEGAN and ESZSL).

| | Baseline | | | Sample probing | | | | | |
| | | | | zsl-loss | | | gzsl-loss | | |
| | u | $s$ | **H** | u | $s$ | **H** | u | s | **H** |
|---|---|---|---|---|---|---|---|---|---|
| CUB | 53.9 | 58.4 | 56.0 | 50.5 | 63.6 | 56.3 | 51.1 | 63.3 | **56.6** |
| FLO | 59.4 | 78.3 | 67.5 | 62.4 | 83.8 | 71.5 | 63.5 | 83.2 | **72.1** |
| SUN | 42.9 | 39.3 | 41.0 | 44.0 | 39.7 | **41.7** | 46.0 | 36.9 | 41.0 |
| AWA | 54.4 | 75.2 | 63.2 | 55.2 | 74.7 | **63.5** | 55.6 | 72.8 | 63.0 |

Table 4: Comparison of mean per-class Fréchet Distance between real and generated unseen class samples on CUB, AWA and FLO datasets for TF-VAEGAN and Our approach. Lower is better.

| | CUB | FLO | AWA |
|---|---|---|---|
| Baseline | 21.5 | 31.0 | 18.5 |
| *Ours* | **19.8** | **30.2** | **17.9** |

## 4.2 ANALYSIS

**ZSL vs GZSL loss in sample probing.** In Table 3, we present a comparison of our approach, using TF-VAEGAN as the generative model and ESZSL as the probe model, when two different types of losses (*zsl-loss* and *gzsl-loss*) are used as $L_{pb}$ in Eq. (3). They differ from each other in terms of classes among which the real examples of probe-validation classes are classified, during the evaluation of sample probing ZSL model. *zsl-loss* and *gzsl-loss* indicate that the examples of probe-validation classes are classified among only probe-validation classes, and both probe-train and probe-validation classes, respectively. We observe that using either one during the evaluation of the sample probing ZSL model, brings its own characteristic results (Table 3). On all datasets, using *zsl-loss* increases the seen accuracy while using *gzsl-loss* increases the unseen accuracy. We choose among these two options using our same hyper-parameter tuning policy, on the validation set.

**Effect of sample probing loss weight.** Figure 3 shows the validation and test set h-score values as a function of sample probing loss weight. In the test set results, we observe an overall increasing performance trend with larger loss weights, up to the weight 6, highlighting the contribution of sample probing. The optimal weight with respect to the validation and the test sets, however, differs. This observation is an example for the difficulty of tuning the ZSL model based on validation set. We set the loss weight to 5 following our hyper-parameter tuning policy, which yields almost 0.5 lower than the maximum test-set score observed for this single hyper-parameter.

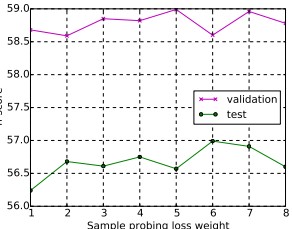

Figure 3: The effect of sample probing weight on the CUB dataset.

**Quantitative analysis of sample quality.** We quantitatively evaluate the sample quality using Fréchet (Wasserstein-2) distance, which is also used in the FID metric for evaluating GAN models. In Table 4, we provide a comparison for TF-VAEGAN and our approach for respective mean per-class Fréchet distances between real and synthetic samples (200 synthetic samples per class) of unseen classes on three datasets. The results show that sample probing helps generator to generate more realistic samples compared to TF-VAEGAN. We further investigate the qualitative analysis of the sample quality via t-SNE visualizations in Appendix B.

## 5 CONCLUSIONS

We propose a principled GZSL approach, which makes use of closed-form ZSL models in generative model training to provide a sample-driven and end-to-end feedback to the generator. Extensive experiments over four benchmark datasets show that the proposed sample probing scheme consistently improves the GZSL results and the sample quality, can easily be integrated into the existing generative GZSL approach and can be utilized with various closed-form probe models.

**Acknowledgement.** This work was supported in part by the TUBITAK Grant 119E597. The numerical calculations were partially performed at TUBITAK ULAKBIM, High Performance and Grid Computing Center (TRUBA resources).

**Reproducibility statement.** We discuss the problem of model selection in GZSL, and the difficulty of making fair comparison across GZSL models in Section 4. We believe that these are open problems in GZSL research. We also clearly explain our hyper-parameter tuning policy, which we use in our experiments. We will provide our source code on a public repository.

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

Table 5: **Comparison against state-of-the-art generative model based GZSL on CUB, FLO, SUN and AWA datasets.** Results obtained with the proposed features are reported, together with the results obtained with fine-tuned features under fine-tuned (FT). The results are reported in terms of top-1 accuracy of unseen (u) and seen (s) classes, together with their harmonic mean (H).

| | | CUB | | | FLO | | | SUN | | | AWA | | |
|---|---|---|---|---|---|---|---|---|---|---|---|---|---|
| | | u | s | **H** | u | s | **H** | u | s | **H** | u | s | **H** |
| | f-CLSWGAN (Xian et al., 2018b) | 3.7 | 57.7 | 49.7 | 59.0 | 73.8 | 65.6 | 42.6 | 36.6 | 39.4 | 57.9 | 61.4 | 59.6 |
| | Cycle-WGAN (Felix et al., 2018) | 47.9 | 59.3 | 53.0 | 61.6 | 69.2 | 65.2 | **47.2** | 33.8 | 39.4 | **59.6** | 63.4 | 59.8 |
| | LisGAN (Li et al., 2019) | 40.9 | 60.5 | 48.8 | 53.1 | 81.7 | 64.4 | 41.5 | 36.6 | 38.9 | 44.2 | 77.0 | 56.1 |
| | f-VAEGAN (Xian et al., 2019) | 48.4 | 60.1 | 53.6 | 56.8 | 74.9 | 64.6 | 45.1 | 38.0 | 41.3 | 57.6 | 70.6 | **63.5** |
| | TF-VAEGAN (Narayan et al., 2020) | 53.9 | 58.4 | 56.0 | 59.4 | 78.3 | 67.5 | 42.9 | 39.3 | 41.0 | 54.4 | 75.2 | 63.2 |
| | Meta-VGAN (Verma et al., 2021) | **55.2** | 48.0 | 53.2 | - | - | - | - | - | - | 57.4 | 70.5 | **63.5** |
| | FREE (Chen et al., 2021) | 51.2 | 61.5 | 55.9 | 62.8 | 80.7 | 70.6 | 46.2 | 37.2 | 41.2 | 48.2 | 78.7 | 59.8 |
| | Ours (based on TF-VAEGAN) | 51.1 | **63.3** | **56.6** | **63.5** | **83.2** | **72.1** | 44.0 | **39.7** | **41.7** | 55.2 | 74.7 | **63.5** |
| | f-VAEGAN (Xian et al., 2019) | 63.2 | 75.6 | 68.9 | - | - | - | **50.1** | 37.8 | 43.1 | **57.1** | 76.1 | **65.2** |
| FT | TF-VAEGAN (Narayan et al., 2020) | **64.2** | 72.7 | 68.2 | 70.0 | 91.3 | 79.2 | 46.5 | **41.7** | **44.0** | 41.7 | **90.2** | 57.0 |
| | Ours (based on TF-VAEGAN) | 63.1 | **76.1** | **69.0** | **70.2** | **91.7** | **79.5** | 47.8 | 40.6 | 43.9 | 45.6 | 87.6 | 60.0 |

Amir R Zamir, Te-Lin Wu, Lin Sun, William B Shen, Bertram E Shi, Jitendra Malik, and Silvio Savarese. Feedback networks. In *Proc. IEEE Conf. Comput. Vis. Pattern Recog.*, pp. 1308–1317, 2017.

Hongyi Zhang, Moustapha Cisse, Yann N Dauphin, and David Lopez-Paz. mixup: Beyond empirical risk minimization. In *Proc. Int. Conf. Learn. Represent.*, 2018.

Yizhe Zhu, Mohamed Elhoseiny, Bingchen Liu, Xi Peng, and Ahmed Elgammal. A Generative Adversarial Approach for Zero-Shot Learning from Noisy Texts. In *Proc. IEEE Conf. Comput. Vis. Pattern Recog.*, pp. 1004–1013, Salt Lake City, UT, USA, June 2018.

## APPENDICES

## A    COMPARISON TO THE STATE-OF-THE-ART

In Table 5, we compare our results with the state-of-the-art generative approaches for GZSL. During training, we select our best model based on the validation results and report test results on models that give best validation scores. For consistency and to keep the baseline comparable to our results, we again report our own results for LisGAN, TF-VAEGAN (without feedback loop) and FREE using our hyper-parameter tuning policy, but do acknowledge that the original papers typically report higher results. The upper part of the table contains results with the original image representations, and the lower part contains those based on fine-tuned representations.

From the results without fine-tuning, we observe that the proposed sample probing based generative model yields state-of-the-art h-scores in all CUB, FLO, SUN and AWA datasets. We also observe competitive results in terms of individual unseen and seen class accuracy values. When compared against results using fine-tuned representations, we again observe state-of-the-art h-scores on CUB and FLO datasets, with a close second on SUN. On AWA, we observe that f-VAEGAN achieves the highest results with a significant margin over our TF-VAEGAN based baseline, where the sample probing improves the baseline yet still achieves a score below that of f-VAEGAN. Overall, while it is hardly fair to compare models with significant implementation details, these results suggest the overall competitiveness of the obtained data generating models with sample probing.

## B    VISUALIZATION OF UNSEEN DATA

To provide additional qualitative insight into the improvements that can be gained using sample probing, we present t-SNE visualizations of synthetic class samples in Figure 4. In the figure, each plot corresponds to an unseen class on the FLO dataset, and the points correspond to the t-SNE embeddings of real samples (× points), generated samples using TF-VAEGAN with sample probing (● points) and those using the baseline TF-VAEGAN model without sample probing (▲ points).

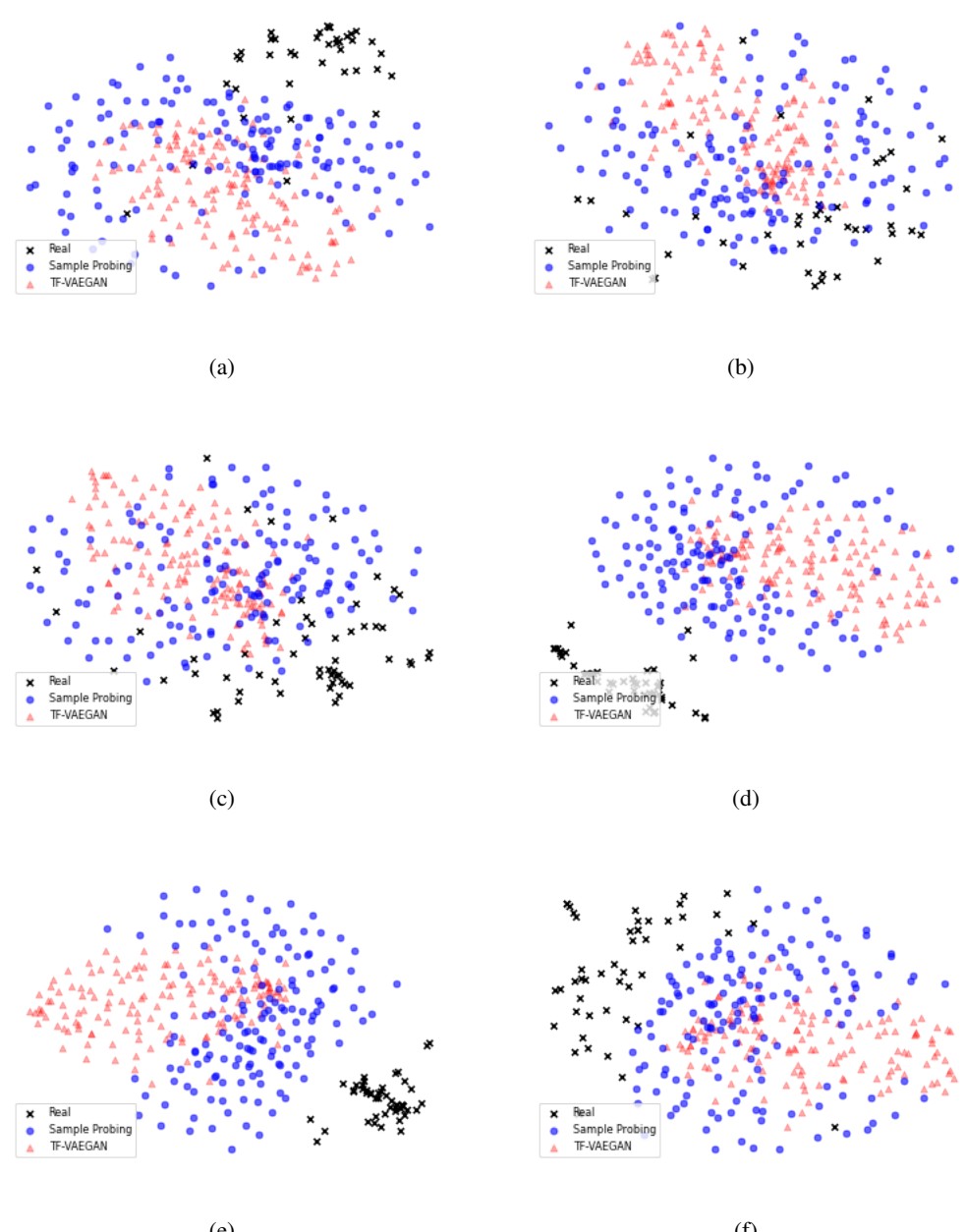

Figure 4: t-SNE visualization of different unseen classes from FLO dataset. Each plot shows t-SNE embeddings of real samples (× points), generated samples using TF-VAEGAN with sample probing (● points) and those using the baseline TF-VAEGAN (Narayan et al., 2020) model without sample probing (▲ points).

From the plots, we can observe that the generative model trained with sample probing tends to yield samples much more aligned with the corresponding true class distributions, compared to those of the baseline model. Overall, these plots demonstrate how sample probing can improve the overall sample quality of a generative model, and possibly lead to superior recognition models when the generated samples are used for classifier training.

