# OpenReview forum: "Closed-form Sample Probing for Learning Generative Models in Zero-shot Learning"
_ICLR.cc/2022/Conference — ICLR 2022 Poster_

### Official Review · Reviewer_yvpj · 2021-11-04

**Correctness:** 3
**Technical Novelty And Significance:** 3
**Empirical Novelty And Significance:** 2
**Recommendation:** 6
**Confidence:** 4

**Main Review:**

This work attempts to address a major concern in current generative ZSL models, i.e., the quality of the synthesized training examples used to train the final GZSL classifier, as the final GZSL performance highly depends on those generated samples. By measuring the quality of those samples during the training process, more informative samples can be obtained, leading to improved performance. Overall, the paper is well-written and easy to follow, with adequate technical details for re-implementation. Below please find the detailed suggestions and questions:

1) In the last paragraph on Page 1, the authors claimed that samples need to be realistic, relevant and informative. However, in the method section, there are no detailed discussions regarding how the proposed solution endows the synthetic samples with these three properties. More clarifications on this point need to be added.

2) Another concern is the performance gain over TF-VAEGAN. From the tables, we can only observe less than 1% increase w.r.t. the harmonic mean in most cases on the benchmarks. One could have expected to see more gains as measuring the sample quality provides important additional information for the generative model.

3) Does the overall framework highly depend on the solvers it adopts? Is it possible to adopt other solvers like the one in SAE [R1]? If employing this solver, can we obtain even more improved GZSL performance as SAE shows the advantage over ESZSL?

4) From Fig. 3, it is interesting to see that the validation set and the test one exhibits the almost opposite trend w.r.t. the harmonic mean. Does this indicate that the hyper-parameter tuning policy is not suitable for the GZSL task?

5) Table 2 only depicts the results on the CUB dataset - how about the performance on the other three datasets?

[R1] Kodirov et al., Semantic Autoencoder for Zero-Shot Learning, CVPR 2017.

**Summary Of The Paper:**

This paper proposes sample probing, a meta-learning scheme for zero-shot learning (ZSL), to measure the quality of the synthetic samples provided by certain generative models. Specifically, an existing closed-form ZSL solver is plugged into an existing generative ZSL framework. Owing to the differentiability of the solver, the whole pipeline is end-to-end trainable. Experiments were conducted on four standard benchmarks, where we can observe the state-of-the-art performance achieved by the proposed sample probing based approach.

**Summary Of The Review:**

This paper addresses an important issue existing in current generative ZSL models. The paper is well-motivated and the solution seems to be effective. The major concern lies in the performance gain and some details regarding the method need further clarifications.

---

> ### Author Response · Authors · 2021-11-23
> **Reply to Reviewer yvpj**
>
> ### Q1. "In the last paragraph on Page 1, the authors claimed that samples need to be realistic, relevant and informative. However, in the method section, there are no detailed discussions regarding how the proposed solution endows the synthetic samples with these three properties. More clarifications on this point need to be added."
>
> Many thanks for pointing out this. To address this shortcoming within the page limits, we have added
>  the following summary and discussion to the end of Section 3: *The proposed
>   approach aims to realize the goal of learning to generate good training samples by evaluating the
>   synthesis quality through the lens of a closed-form trainable probe model, the prediction loss of
>   which is used as a loss for the G updates. Therefore, G is expected to be progressively guided
>   towards producing realistic, relevant and informative samples, through the reinforcement of which
>   may vary depending on the inherent nature of the chosen probe model.*
>
>
> ### Q2. "Another concern is the performance gain over TF-VAEGAN. From the tables, we can only observe less than 1% increase w.r.t. the harmonic mean in most cases on the benchmarks. One could have expected to see more gains as measuring the sample quality provides important additional information for the generative model."
>
> The revised manuscript provides a much stronger validation of the proposed
> approach. In particular, the greatly expanded results in Table 1 now demonstrate that sample probing
> improves the results in 17 out of 19 cases
> (using sample probing with the cWGAN, LisGAN,
> TF-VAEGAN, TF-VAEGAN with fine-tuned features and FREE baselines, on 4 benchmark datasets),
> with improvements up to 4.6 points. We similarly now evaluate the proposed approach with multiple
> closed-form probe models (Table 2). Overall, the improvements over already
> strong baselines validate the effectiveness of the proposed sample probing
> approach.
>
>
> ### Q3. "Does the overall framework highly depend on the solvers it adopts? Is it possible to adopt other solvers like the one in SAE [R1]? If employing this solver, can we obtain even more improved GZSL performance as SAE shows the advantage over ESZSL? ([R1] Kodirov et al., Semantic Autoencoder for Zero-Shot Learning, CVPR 2017.)"
>
> Many thanks for this suggestion. Upon this comment, we looked into alternative
> closed-form probe models, starting with SAE. It is not immediately clear
> whether the full SAE model can be used for this purpose, as its training
> requires the solution of a Sylvester equation.  We instead adapted the Vis2Sem
> model from the SAE paper, and Sem2Vis model from Shigeto et al. (ECML/PKDD
> 2015) as probe models. The results are reported in the newly added Table 2. We
> observe that using the sample probing with alternative probe models performs
> equivalent to or better than the baseline in 11 out of 12 cases. In addition,
> we also observe that Sem2Vis performs the best on CUB (+0.9 over the baseline),
> ESZSL provides a clear gain on FLO (+4.6) and a relative improvement on AWA
> (+0.3), and Vis2Sem improves the most on SUN (+1.8). These results suggest that
> sample probe alternatives have their own advantages and disadvantages.
>
>
> ### Q4. "From Fig. 3, it is interesting to see that the validation set and the test one exhibits the almost opposite trend w.r.t. the harmonic mean. Does this indicate that the hyper-parameter tuning policy is not suitable for the GZSL task?"
>
> We believe that the difficulty of setting hyper-parameters in (generative) GZSL
> is a general problem that is not specific to a particular model or model
> selection policy, due to sharp differences in terms of the class distributions
> observed during training and validation versus testing.
>
> ### Q5. "Table 2 only depicts the results on the CUB dataset - how about the performance on the other three datasets?"
>
> We believe that the revised Table 1 and the new Table 2 addresses this problem by providing a much more comprehensive
> evaluation of the approach, as explained above for the preceding questions Q2 and Q3.

---

### Official Review · Reviewer_NR3H · 2021-11-06

**Correctness:** 3
**Technical Novelty And Significance:** 3
**Empirical Novelty And Significance:** 2
**Recommendation:** 6
**Confidence:** 3

**Main Review:**

Paper strengths:

1) The paper is well written and easy to follow and understand.
2) The motivations for the work are clear, as is the method itself.
3) Providing greater detail on hyperparameter choices is great, and as the authors demonstrate - it is also crucial.
4) The proposed method is also general in the sense that it can be readily applied to future work. In my opinion, this is a major selling point of the work.
5) Results appear to indicate an improvement over the state of the art.

Note, however, see point (1) in the weaknesses section in regards to points 4 and 5.

Paper weaknesses:

While the method is promising and the suggested approach makes intuitive sense, I feel like the experimental results do not currently support it well enough.

In particular (listed in order of importance for my evaluation):

1)

As the author's note in their closing remarks: "Our method works in an end-to-end manner and it can
be easily integrated into any mainstream generative zero-shot learning framework.". This is, in my opinion, one of the most significant selling points of the paper. Alas, it is not investigated. The method is applied only to a single baseline model*, where it shows improved performance only when the baseline is trained with a different set of hyper-parameters than in it's original implementation.

*There is an additional experiment on a single dataset (out of 4) with a different, very basic baseline.

1.1) What were your results when you used the same hyper-parameters as the baseline model? Why do you think your model outperforms the baseline with a specific number of iterations, but underperforms with another? Perhaps your model is simply more efficient and converges faster, but introduces other problems which mitigate the advantage in the long-run?

1.2) If things are highly hyper-parameter sensitive, why did you optimize only their number of iterations? What about the other parameters?

1.3) Can your method be applied to other generative ZSL models? I would have more confidence in the general applicability of the method if it demonstrated improvements when integrated into multiple existing works.

In this context, Table 3 (where most of the results reside) is largely irrelevant. Most of these numbers are simply an indication that TF-VAEGAN is better than its competition. (I am not advocating the table's removal - but applying your method to the models listed there would be a considerable improvement).

2)

Do you have any intuition as to why the choice of ZSL / GZSL in your loss (i.e. table 4) is so crucial that a wrong choice may make your model perform equally to, or worse than the baseline on some datasets? Is this merely a function of how well the original model performs on seen vs unseen classes? (this is something that could be seen if we had more adapted baselines to compare with!)

3)

How much of an effect does your method have on training times? Does the closed-form solution of EZSL have a noticeable impact on the time required per training iteration?

4)

A natural alternative to your approach may be the use of meta-learned ZSL model in place of the closed-form solution. This would allow updating the model with a single training iteration which may produce sufficiently strong gradients for your generator. My knowledge of recent works in the field is limited, are you aware of any works doing something similar? If so, comparisons to them could strengthen your work.

5)

Given the aim of increasing reproducibility by reporting hyper-parameter tuning methodologies, I would add answers to the following questions (at the very least to the released code):

5.1) Are you using the same 20% validation split for all experiments on a given dataset? If so, can this split be released?

5.2) Did you use a single split, or cross-validate?

6)

Sample quality metrics - Fréchet distance is typically sensitive to the number of generated samples used in the comparison. These should be reported to facilitate future comparisons.


**Summary Of The Paper:**

The paper outlines a method for improving generative zero-shot learning (ZSL) approaches. Rather than simply training the generative model with the goal of reproducing some real data distribution, the work proposes to train it with an additional goal of synthesizing samples that directly improve the performance of the downstream classification model.

The authors propose to do so through a novel sample-probing loss in which generated samples are used to train a closed-form ZSL model with a differentiable solution. The ZSL model is then evaluated directly on the classification task - and gradients are back-propagated to the generative model's parameters. By applying this approach to an existing generative ZSL model, the authors demonstrate improved sample quality for their synthetic data and increased classification performance across multiple datasets.

In summary, the paper's contributions are:
1) A major contribution: An approach to improving existing generative ZSL methods with a loss that maximizes their performance on the downstream task.
2) A minor contribution: A more detailed and rigorous reporting of the methodologies used to fine-tune model hyperparameters, aimed at increasing reproducibility.

**Summary Of The Review:**

Overall, I enjoyed the paper and would have liked to recommend acceptance. The approach makes intuitive sense, and can no doubt be extended with multiple future works, offering the community a parallel line of investigation into improving GZSL results.

However, I think the flaws outlined in point (1) of the weakness section are fundamental enough that I am worried that any future works which try to build on this one might simply be wasting their time. I do not believe this to be the case, but I would like to see additional experimental results that would convince me otherwise.


*I have marked my confidence as 3 due to limited familiarity with related work. As an extension to that, I may have missed prior art which already suggested similar ideas.

******************************
Post rebuttal update:

The authors conducted an extensive set of experiments and addressed my primary concerns (the method's generalizability to additional baselines). I am still concerned about some aspects of the evaluation (considerably worse results for all baselines compared to their originally reported values). However, since the paper suggests a method for improving other models, the relative improvements are what matters most. As such, I am willing to accept the current demonstration of a (fairly) consistent improvement when using the same hyper-parameter selection approach across the board.

Even if accepted, I urge the authors to better highlight and explain the difference between their experimental results and the original baseline values.

---

> ### Author Response · Authors · 2021-11-23
> **Reply to Reviewer NR3H**
>
>
> ### Q1. "As the author's note in their closing remarks: "Our method works in an end-to-end manner and it can be easily integrated into any mainstream generative zero-shot learning framework.". This is, in my opinion, one of the most significant selling points of the paper. Alas, it is not investigated. The method is applied only to a single baseline model\*, where it shows improved performance only when the baseline is trained with a different set of hyper-parameters than in it's original implementation. (\* There is an additional experiment on a single dataset (out of 4) with a different, very basic baseline.)"
>
> Many thanks for this comment. Upon suggestion, we have additionally evaluated
> our model with two additional generative models and also expanded the cWGAN
> experiments.  As a result, the greatly extended Table 1 now reports results by
> evaluating sample probing with 5 generative GZSL models (cWGAN, LisGAN,
> TF-VAEGAN, TF-VAEGAN with fine-tuned features and FREE), on 4 benchmark
> datasets.  We observe that sample probing improves the results in 17 out of 19
> cases, with improvements up to 4.6 points in some cases.
>
> ### Q2. "What were your results when you used the same hyper-parameters as the baseline model? Why do you think your model outperforms the baseline with a specific number of iterations, but underperforms with another? Perhaps your model is simply more efficient and converges faster, but introduces other problems which mitigate the advantage in the long-run?"
>
> We believe that the difficulty of setting hyper-parameters in (generative) GZSL
> is a general problem that is not specific to a particular model, as we
> discuss in Section 4.  In our own experience, we observe that the number of
> training iterations is particularly important as it directly controls the degree of (over)-fitting to the seen classes.
> For example, using the TF-VAEGAN
> model with sample probing on the FLO dataset, we have repeated the experiment
> with 40 random hyper-parameter combinations.  If we set the number of training
> iterations using the proposed policy (i.e. based on validation h-score), the
> test set h-score varies between 65.6 and 72.7 (We report the result with 72.1
> test h-score, using our hyper-parameter tuning policy). However, if we were to set only
> the number of training generative model iterations with respect to the test set h-score,
> the test set h-score would vary between 72.2 and 73.7, at a greatly improved and narrow range!
> On the same dataset, the original number of training iterations of the official source code yields 69.8 as the test set h-score.
>
> These observations suggest that the main (and possibly the only) clearly
> sensitive hyper-parameter is the number of training iterations, possibly not
> for reasons specific to the proposed sample probing scheme, but instead the
> nature of (G)ZSL where the test set includes samples of completely different
> classes.  In addition, we emphasize that simple performance-based comparisons across papers can
> be misleading due to a variety of implementation differences, such as the difference in the way
> the number of training iterations hyper-parameter is tuned.
>
>
> ### Q3. "If things are highly hyper-parameter sensitive, why did you optimize only their number of iterations? What about the other parameters?"
>
> We believe that our answer to the preceding question explains our focus on the
> number of iterations. There are two reasons why we opt to keep the remaining
> hyper-parameters fixed: (i) Our focus is the evaluation of the proposed
> approach in terms of the relative performance with and without sample probing,
> using strong generative GZSL baselines, not the evaluation of the baselines
> themselves. Since we need to tune the number of training iterations upon the
> inclusion of our new loss term, we only tune this hyper-parameter for the
> baseline models using our model selection policy to make the results
> comparable.  (ii) Most baseline models contain a large number of
> hyper-parameters, re-tuning all of them is not feasible with our compute
> resources.
>
> ### Q4. "Can your method be applied to other generative ZSL models? I would have more confidence in the general applicability of the method if it demonstrated improvements when integrated into multiple existing works."
>
> Many thanks for this suggestion. In the revised manuscript, we now evaluate the
> proposed approach using two additional closed-form ZSL models as the probe
> models, and report the results in the newly added Table 2.  As discussed in
> Section 4.1, we observe improvements via sample probing in 11 out of 12 cases,
> which supports the general applicability of the method.

---

> > ### Comment · Reviewer_NR3H · 2021-11-24
> > **Post Rebuttal Update**
> >
> > Thank you for the comprehensive response and for the updates to the manuscript.
> >
> > I have gone over the replies to all reviewers, and over the changes to the paper.
> >
> > My main concerns seem to be well addressed. However, I am confused about the fact that all baseline methods seem to have lower scores than reported in their respective papers / in other works which cite them. In many cases, these gaps are larger than the advantage that your method brings. Is this a fundamental reproducibility issue? If these papers provided code, then surely their hyperparameters are available and this is not an issue of missing information (I can see for example that FREE's official implementation does contain such details). I understand that you may not know how these parameters were chosen, but why does this preclude you from applying your method over the baseline with the same parameters?
> >
> > I am tentatively increasing my score to a 6 accordingly. However, I would like to see a response to this issue.
> >
> > Two minor notes -
> >
> > 1) I am also not a big fan of claiming a 0.1 advantage as an improvement (LisGAN on AWA). It's extremely likely that this gap is within the 1-sigma confidence interval for re-training with random seeds (for ImageNet classification top-1 accuracies these are typically ~0.2-0.5% depending on the underlying architecture).
> >
> > 2) Appendix B: What are the classes? Moreover, I feel like the claim that sample probing tends to yield samples much more aligned with the corresponding true class distributions is overstated. In some cases (a, e) I could potentially accept this, but for others (b, d, f) your method also generates samples which are worse than the baseline, with the highest concentration of samples seemingly aligned between your method and the baseline.

---

> ### Author Response · Authors · 2021-11-23
> **Reply to Reviewer NR3H (continued)**
>
>
> ### Q5. "In this context, Table 3 (where most of the results reside) is largely irrelevant. Most of these numbers are simply an indication that TF-VAEGAN is better than its competition. (I am not advocating the table's removal - but applying your method to the models listed there would be a considerable improvement)."
>
> We agree that Table 5 in Appendix A (previously Table 3 in the main text) provides only a coarse comparison. We try to emphasize this point in the paper.
>
> ### Q6. "Do you have any intuition as to why the choice of ZSL / GZSL in your loss (i.e. table 4) is so crucial that a wrong choice may make your model perform equally to, or worse than the baseline on some datasets? Is this merely a function of how well the original model performs on seen vs unseen classes? (this is something that could be seen if we had more adapted baselines to compare with!)"
>
> Thank you for the interesting question.  To better present the results regarding this choice,
> we first have added the baseline results to Table 3. From these results, we observe that
> even when the sub-optimal version of the sample probe loss is used, the final performance is
> typically as good as or better than the baseline, with only exception being gzsl-loss on AWA.
>
> In addition, to provide additional insight into the chose of loss, we report the version of the loss
> chosen using our model selection protocol in the table below:
>
> |              | CUB  | FLO  | SUN | AWA  |
> |--------------|------|------|-----|------|
> | cWGAN        | gzsl | gzsl | zsl | -    |
> | LisGAN       | gzsl | gzsl | zsl | zsl  |
> | FREE         | gzsl | gzsl | zsl | zsl  |
> | TF-VAEGAN    | gzsl | gzsl | zsl | zsl  |
> | TF-VAEGAN-FT | gzsl | zsl  | zsl | gzsl |
>
> This table reveals an interesting pattern: independent from the generative model being used, a single
> version of the loss tends to be selected in each dataset. The only exceptions are observed with the
> fine-tuned features, which is not unusual considering that the visual data is (almost) completely
> different. This pattern highlights that this choice is data dependent, most likely due to various
> non-trivial factors.  Therefore, overall, we think that this choice needs to be made in
> a data-driven way, via model selection.
>
> ### Q7. "How much of an effect does your method have on training times? Does the closed-form solution of EZSL have a noticeable impact on the time required per training iteration?"
>
> As the sample probing model involves using a closed-form solver at each single generative model update step, it does have an extra cost.  For example, training TF-VAEGAN for 100 epochs on the CUB dataset takes 27.2 minutes without sample probing, and 43.2 minutes with sample probing.
> Despite its overhead, since all the experiments are in the feature domain as in other mainstream generative GZSL studies, the overall experiment durations remain practically feasible, in the order of 1 to 5 hours (depending on the maximum number of iterations, and the dataset size).
>
> ### Q8. "A natural alternative to your approach may be the use of meta-learned ZSL model in place of the closed-form solution. This would allow updating the model with a single training iteration which may produce sufficiently strong gradients for your generator. My knowledge of recent works in the field is limited, are you aware of any works doing something similar? If so, comparisons to them could strengthen your work."
>
> We are not aware of a generative GZSL work similar to the one described in the question, or a plug-and-play meta-learning based GZSL model alternative for our closed-form solvers. However, as it points to a very different approach that needs to be studied in detail, we think this is an interesting future work direction that is beyond the scope of our work. We also believe the inclusion of alternative closed-form ZSL models also improves the evaluation of the proposed approach in a similar direction.
>
> ### Q9. "Are you using the same 20\% validation split for all experiments on a given dataset? If so, can this split be released?"
>
> We use the same 20\% validation split on all experiments of each dataset. In addition, although each dataset's validation splits are randomly constructed, we have ensured that these splits are fully reproducible. We will share all split details upon publication.
>
> ### Q10. "Did you use a single split, or cross-validate?"
>
> We generate a single validation split per dataset.
>
> ### Q11. "Sample quality metrics - Fréchet distance is typically sensitive to the number of generated samples used in the comparison. These should be reported to facilitate future comparisons."
>
> In Section 4.2, we have added the explanation denoting that we use 200 synthetic samples per class, and all real
> samples of the unseen classes. We have not observed any major changes in the results depending on the
> number of synthetic samples.

---

### Official Review · Reviewer_dVi5 · 2021-11-08

**Correctness:** 3
**Technical Novelty And Significance:** 2
**Empirical Novelty And Significance:** 2
**Recommendation:** 5
**Confidence:** 2

**Main Review:**

Strengths:

1- The paper is well written.

2- This approach tries to generate realistic, relevant, and informative features train an accurate classifier using a sample probing mechanism.

3- It also uses feature fine-tune to improve the performance further.

Weaknesses:

1.  The novelty of the proposed approach is limited. The main contribution of this paper is the sample probing technique to generate features. However, prior works ([1],[2],[3]) have proposed a very similar approach. Therefore, given [1],[2],[3] methods, the contribution of this paper is not novel.

2. Considering prior works ([3], and other results from Table 1 of [3]), the accuracy improvements are either not significant or are lower.


[1] Episode-Based Prototype Generating Network for Zero-Shot Learning

[2] Meta-Learning for Generalized Zero-Shot Learning

[3] Meta-Learned Attribute Self-Gating for Continual Generalized Zero-Shot Learning

**Summary Of The Paper:**

This paper proposes a generalized zero-shot learning approach by providing feedback for generator with the evaluation of real samples of unseen classes.

**Summary Of The Review:**

Considering the weaknesses of the paper, I would give this paper marginally below the acceptance threshold.

---

> ### Author Response · Authors · 2021-11-23
> **Reply to Reviewer dVi5**
>
> ### Q1. "The novelty of the proposed approach is limited. The main contribution of this paper is the sample probing technique to generate features. However, prior works ([1],[2],[3]) have proposed a very similar approach. Therefore, given [1],[2],[3] methods, the contribution of this paper is not novel. ([1] Episode-Based Prototype Generating Network for Zero-Shot Learning[2] Meta-Learning for Generalized Zero-Shot Learning [3] Meta-Learned Attribute Self-Gating for Continual Generalized Zero-Shot Learning)"
>
> We believe that the revised manuscript much better presents the novelty of the proposed approach as a
> general technique to improve generative GZSL model training. Most notable changes that
> better highlight the contribution:
> * Much more comprehensive evaluation of the approach over multiple generative models (Table 1).
> * Evaluation of the approach's usability with alternative probe models (Table 2), demonstrating the
>   generality of the approach.
> * The following method summary and discussion is added to the end of Section 3: *The proposed
>   approach aims to realize the goal of learning to generate good training samples by evaluating the
>   synthesis quality through the lens of a closed-form trainable probe model, the prediction loss of
>   which is used as a loss for the G updates. Therefore, G is expected to be progressively guided
>   towards producing realistic, relevant and informative samples, through the reinforcement of which
>   may vary depending on the inherent nature of the chosen probe model.*
>
> We believe that the aforementioned improvements and explanations will better highlight the
> distinction of our work. We also note that the closed-form probe model is not a part of [1],[2],[3], and
> overall our work significantly differs from them. The main differences can be summarized as follows:
>
> * [1] (Yu et al Episode-Based Prototype Generating Network for Zero-Shot Learning, CVPR 2020): this
>   work differs categorically from ours as it does not aim to train a generative model for sample
>   generation. The prototype generation refers to learning a deterministic mapping from attributes to
>   class prototypes, much like the attribute-to-visual mapping of a bi-linear compatibility function.
>   Therefore, we do not see any close relation to ours.
>
> * [2] (Verma et al., Meta-Learning for Generalized Zero-Shot Learning, AAAI 2020) is a pioneering work
>   on the use of meta-learning principles for generative GZSL model training.  However, this work
>   follows a different direction by integrating Model-Agnostic Meta-Learning (MAML) update rules.  At
>   each step, a single-step update is first tentatively applied to the model components over real
>   seen class training examples as the inner-loop update, and the loss of the updated components is
>   then computed over real unseen class samples. The resulting compute graph is then used to make a
>   permanent gradient descent/ascent update step. In contrast, at each step, our proposed approach
>   samples a batch of synthetic examples from the generative model, **fully trains** a tentative ZSL
>   model using the synthetic examples, and computes the tentative ZSL model's loss on novel real examples
>   to be used as a loss for updating the generative model. The complete training of a tentative model
>   per step and end-to-end differentiability of the whole compute chain is made possible by the idea
>   of leveraging closed-form solvable formulations as the tentative sample probe models, which has
>   not been explored in any prior work that we are aware of.
>   Therefore, the technical contribution of [2] is fundamentally different from ours.
>
> * [3] (Verma et al., Meta-Learned Attribute Self-Gating for Continual Generalized Zero-Shot Learning, arXiv 2021):
>   We weren't aware of this manuscript, which appears to be an arXiv-only manuscript currently. The work focuses
>   on a different problem (continual zero-shot learning) and proposes a non-generative approach. Since our main contribution
>   is a novel technique for improving generative GZSL models, we believe that there is no close connection between [3] and our work.
>
> Finally, we also note that Section 2 provides a detailed discussion of [1] and
> [2] (the arXiv manuscript [3] is omitted due to lack of space, and overall weak
> relation).

---

> ### Author Response · Authors · 2021-11-23
> **Reply to Reviewer dVi5 (continued)**
>
> ### Q2. "Considering prior works ([3], and other results from Table 1 of [3]), the accuracy improvements are either not significant or are lower."
>
> The revised manuscript provides a much stronger validation of the proposed
> approach. In particular, the greatly expanded results in Table 1 now
> demonstrate that sample probing improves the results in 17 out of 19 cases
> (using sample probing with the cWGAN, LisGAN,
> TF-VAEGAN, TF-VAEGAN with fine-tuned features and FREE baselines, on 4 benchmark datasets), with
> improvements up to 4.6 points. We similarly now evaluate the
> proposed approach with multiple closed-form probe models (Table 2). Overall,
> the improvements over already strong baselines validate the effectiveness of
> the proposed sample probing approach.  The newly added t-SNE visualizations
> (Appendix B) complements the Fréchet Distance results and provides further
> insight into the positive impact of sample probing.

---

### Official Review · Reviewer_UDYK · 2021-11-08

**Correctness:** 3
**Technical Novelty And Significance:** 2
**Empirical Novelty And Significance:** 2
**Recommendation:** 6
**Confidence:** 4

**Main Review:**

Paper Strengths：

The authors tackle an important and challenging problem of generative modeling based generalized zero-shot learning. The proposed approach is simple. Experimental evaluations demonstrate the effect by introducing the sample probing method and end-to-end training.

Paper Weaknesses：

1) In the related work section, the authors discussed meta-learning and few-shot classification. However, there was a lack of discussion on using generative models to perform data augmentation for meta-learning and few-shot classification, such as [Low-shot visual recognition by shrinking and hallucinating features, ICCV, 2017] [MetaGAN: An adversarial approach to few-shot learning, NeurIPS, 2018] [Low-shot learning from imaginary data, CVPR, 2018] [Delta-encoder: an effective sample synthesis method for few-shot object recognition, NeurIPS, 2018] [Image deformation meta-networks for one-shot learning, CVPR, 2019]. While zero-shot and few-shot learning are different, they are related. In particular, I feel that the generative model framework in this paper is similar to [Low-shot learning from imaginary data, CVPR, 2018], where the classification loss is used to train the generator in an end-to-end fashion and the generator is integrated into the meta-training framework. An in-depth discussion on this is needed.

2) The proposed approach relies on a closed-form zero-shot model, which is also the main technical contribution of this paper. For this purpose, the ESZSL model [Romera-Paredes & Torr, 2015] is used. However, it seems that such requirement makes the method restrictive to this zero-shot model, and thus is not general and cannot be combined with more state-of-the-art zero-shot models. Even for ESZSL [Romera-Paredes & Torr, 2015], its closed-form solution only applies for some particular parameter setting.

3) From the results, like in Table 1 and Table 3, the improvements of the proposed approach seem quite marginal, especially when the features are fine-tuned. Also, the proposed approach is not consistently better than existing methods. For example, in Table 3, for the fine-tuning setting, on 2 out of 4 benchmarks, the proposed approach is worse than the state of the art.

4) It would be interesting to see the ablation study regarding the different loss components.

5) Somehow it is a little vague – is the paper synthesizing raw images or feature vectors in the pre-trained feature space? This is only explicitly mentioned until it talks about the implementation details. It would be better to make this explicit starting in the introduction.

6) Following the previous comment, it would be interesting to visualize the synthesized features, like using t-SNE visualizations, and analyze why they are helpful.

7) There are a bunch of hyper-parameters involved in the proposed approach. The authors also mentioned the difficulty in consistently setting up these hyper-parameters. Moreover, how is the hyper-parameter sensitivity?

8) There are several grammar issues and typos in the paper. Please proofread.

**Summary Of The Paper:**

This paper aims to address the problem of generalized zero-shot classification based on generative models. The main contribution is that it considers training and evaluating a generative model in synthesizing training examples that are helpful to improve classification performance. To this end, it leverages the zero-shot learning model of ESZSL that can be efficiently fit using a closed-form solution. This then serves as a sample probing mechanism, so that the generator receives training signal directly based on the value of its samples for classifier training and thus enables end-to-end training. The corresponding sample probing loss function is added into the standard generative model training loss. The final training procedure is performed in a way that is similar to meta-training. The approach is tested on standard generalized zero-shot classification setups, including CUB, SUN, AWA2, and FLO, and compared with state-of-the-art results.

**Summary Of The Review:**

The paper introduces sample probing to train a generative model that synthesizes data in feature space for generalized zero-shot learning. The proposed approach is restrictive to certain zero-shot learning models. Some ablation studies and analysis are missing.

=================

post rebuttal:

I thank the authors for the extensive experiments and clarification made in the rebuttal. The rebuttal has addressed most of my concerns, especially showing the generalizability of the proposed approach (different types of generative models and different types of closed-form ZSL models), so I raised my score. However, I am still a little concerned about the novelty of the proposed approach and its marginal improvements.

---

> ### Author Response · Authors · 2021-11-23
> **Reply to Reviewer UDYK**
>
> ### Q1. "In the related work section, the authors discussed meta-learning and few-shot classification. However, there was a lack of discussion on using generative models to perform data augmentation for meta-learning and few-shot classification, such as [Low-shot visual recognition by shrinking and hallucinating features, ICCV, 2017] [MetaGAN: An adversarial approach to few-shot learning, NeurIPS, 2018] [Low-shot learning from imaginary data, CVPR, 2018] [Delta-encoder: an effective sample synthesis method for few-shot object recognition, NeurIPS, 2018] [Image deformation meta-networks for one-shot learning, CVPR, 2019]. While zero-shot and few-shot learning are different, they are related. In particular, I feel that the generative model framework in this paper is similar to [Low-shot learning from imaginary data, CVPR, 2018], where the classification loss is used to train the generator in an end-to-end fashion and the generator is integrated into the meta-training framework. An in-depth discussion on this is needed."
>
> Many thanks for pointing this out. We agree that despite the clear differences across GZSL and few-shot
> learning, the generative approaches to both problems naturally have similar motivations and
> technical similarities. We also agree that the relation to "Low-shot learning from imaginary data" needs to be clarified, which we had overlooked.
> For this reason, we have added the following discussion to Section 2: "Another related research topic is generative few-shot learning (FSL), where the goal is learning to synthesize additional in-class samples from few examples, e.g. Hariharan & Girshick (2016); Wang et al. (2018); Gao et al. (2018); Schwartz et al. (2018); Lazarou et al. (2021). Among these, Wang et al. (2018) is particularly related for following a similar motivation of learning to generate good training examples. This is realized by feeding generated samples to a meta-learning model to obtain a classifier, apply it to real query samples, and use its query loss to update the generative model. Apart from the main difference in the problem definition (GZSL vs FSL), our work differs mainly by fully-training a closed-form solvable ZSL model from scratch at each training step, instead of a few-shot meta-learners that is jointly trained progressively with the generative model.*
>
> We also note that it is not immediately clear how the approach in
> *Low-shot learning from imaginary data* shall be adapted to GZSL, which
> is beyond the scope of our work. We think the in-depth exploration of
> the relative advantages/disadvantages of using progressively updated meta-learners (trained jointly with the generative
> model throughout training)
> versus our approach of closed-form solvable probe models (re-trained fully from scratch on each
> synthetic sample batch)
> can be an interesting future work direction.
>
> ### Q2. "The proposed approach relies on a closed-form zero-shot model, which is also the main technical contribution of this paper. For this purpose, the ESZSL model [Romera-Paredes & Torr, 2015] is used. However, it seems that such a requirement makes the method restrictive to this zero-shot model, and thus is not general and cannot be combined with more state-of-the-art zero-shot models. Even for ESZSL [Romera-Paredes & Torr, 2015], its closed-form solution only applies for some particular parameter setting."
>
> While the exploitation of closed-form ZSL models for sample probing has a
> central role in our approach, the requirement of using closed-form ZSL models can
> indeed be seen as a limitation as well. We believe, however, the revised manuscript
> now experimentally demonstrates that our approach is not restricted to ESZSL, via the
> alternative probe model experiments presented in Table 2, and discussed in Section 4.1.
>
>
> ### Q3. "From the results, like in Table 1 and Table 3, the improvements of the proposed approach seem quite marginal, especially when the features are fine-tuned. Also, the proposed approach is not consistently better than existing methods. For example, in Table 3, for the fine-tuning setting, on 2 out of 4 benchmarks, the proposed approach is worse than the state of the art."
>
> The revised manuscript provides a much stronger validation of the proposed
> approach. In particular, the greatly expanded results in Table 1 now demonstrate that sample probing
> improves the results in 17 out of 19 cases
> (using sample probing with the cWGAN, LisGAN,
> TF-VAEGAN, TF-VAEGAN with fine-tuned features and FREE baselines, 4 benchmark datasets), with improvements up to 4.6 points in
> some cases. We similarly now evaluate the proposed approach with multiple
> closed-form probe models (Table 2). Overall, the improvements over already
> strong baselines validate the effectiveness of the proposed sample probing
> approach.  The newly added t-SNE visualizations (Appendix B) complements the
> Fréchet Distance results and provides further insight into the positive impact
> of sample probing.

---

> ### Author Response · Authors · 2021-11-23
> **Reply to Reviewer UDYK (continued)**
>
> ### Q4. "It would be interesting to see the ablation study regarding the different loss components."
>
> We believe that the revised manuscript also presents an improved evaluation of
> loss components. In addition to the already existing ZSL versus GZSL loss for
> sample probing and evaluation of the effect of sample probing loss weight, we
> now also report the effect of using alternative probe models.
>
> ### Q5. "Somehow it is a little vague – is the paper synthesizing raw images or feature vectors in -the pre-trained feature space? This is only explicitly mentioned until it talks about the implementation details. It would be better to make this explicit starting in the introduction."
>
> Many thanks for pointing this out. We indeed follow the mainstream generative GZSL works and focus
> on the feature representations obtained from a pre-trained model. We have now clarified this in
> Section 3, with the following revised sentence: *In our work, we focus on ZSL models where $\X$ is
> the space of image representations extracted using a pre-trained ConvNet.*
>
> ### Q6. "Following the previous comment, it would be interesting to visualize the synthesized features, like using t-SNE visualizations, and analyze why they are helpful."
>
> We are thankful for this suggestion. We have added t-SNE visualizations and a
> related discussion to Appendix B. The t-SNE visualizations suggest that the
> generative model trained with sample probing tends to yield samples much more
> aligned with the corresponding true class distributions, compared to those of
> the baseline model.
>
> ### Q7. "There are a bunch of hyper-parameters involved in the proposed approach. The authors also mentioned the difficulty in consistently setting up these hyper-parameters. Moreover, how is the hyper-parameter sensitivity?"
>
> We believe that the difficulty of setting hyper-parameters in (generative) GZSL
> is a general problem that is not specific to a particular model, as also
> discussed in Section 4.  In our own experience, we observe that the number of
> training iterations, which directly controls the degree of (over)-fitting to seen classes, is
> the single most difficult-to-tune hyper-parameter. For example, using the TF-VAEGAN
> model with sample probing on the FLO dataset, we have repeated the experiment
> with 40 random hyper-parameter combinations.  If we set the number of training
> iterations using the proposed policy (i.e. based on validation h-score), the
> test set h-score varies between 65.6 and 72.7 (We report the result with 72.1
> test h-score, using our hyper-parameter tuning policy). However, if we were to set only
> the number of training generative model iterations with respect to the test set h-score,
> the test set h-score would vary between 72.2 and 73.7, at a higher and narrower range!
>
> These observations suggest that (i) the main (and possibly the only) clearly sensitive hyper-parameter
> is the number of training iterations, (ii) simple performance-based comparisons across papers can
> be misleading due to a variety of implementation differences, such as the difference in the way
> the number of training iterations hyper-parameter is tuned.
>
>
> ### Q8. "There are several grammar issues and typos in the paper. Please proofread."
>
> Many thanks, again, for your careful review.  We have carefully proofread the
> paper, fixed all the problems that we could find.

---

### Official Review · Reviewer_vUDu · 2021-11-09

**Correctness:** 4
**Technical Novelty And Significance:** 2
**Empirical Novelty And Significance:** 2
**Recommendation:** 5
**Confidence:** 5

**Main Review:**

Strong Points:  1- The presentation is clear, and the research problem is well motivated.
2- The proposed method is evaluated on four relatively comprehensive benchmark datasets.


Weaknesses: 1- The paper could not highlight its novelty well. The idea to improve model generalization ability with cross-validation is not new. The proposed method seems to be an integration of generative models and the existing closed-form solution.  The closed-form probe model is borrowed from [a],[b],[c].

2- To prove the better efficacy of the proposed model, it should be trained using a few examples (5/10 samples) per seen class.

3- The experiments for a large dataset (ImageNet) should be included for better evaluation.


[a]- Meta-Learning for Generalized Zero-Shot Learning, AAAI 2020.
[b]- Episode-Based Prototype Generating Network for Zero-Shot Learning, CVPR 2020.
[c]- Towards Zero-Shot Learning With Fewer Seen Class Examples, WACV 2021.


**Summary Of The Paper:**


The paper addresses the task of GZSL – more specifically, they provide a way to improve the quality of the generative samples in generative GZSL. A closed-form probe model is introduced to provide an efficient and differentiable solution in compute graph. In this manner, the generator receives feedback directly based on the value of its samples for model training purposes. It shows the results on two different settings, with fine-tuning features and without fine-tuning the features.



**Summary Of The Review:**


Due to lack of novelty and the performance of the proposed model is marginally above the existing approaches.

---

> ### Author Response · Authors · 2021-11-23
> **Reply to Reviewer vUDu**
>
> ### Q1. "The paper could not highlight its novelty well. The idea to improve model generalization ability with cross-validation is not new. The proposed method seems to be an integration of generative models and the existing closed-form solution."
>
> We believe that our work's resemblance to cross-validation is only motivational, otherwise, the
> proposed sample probing scheme for generative model training versus cross-validation has no close
> technical relation. In particular, cross-validation is a simple train and test based model
> evaluation model, which is also used for model selection in a non-differentiable way. Instead, we
> define a novel end-to-end differentiable approach to guide conditional generative model training for GZSL purposes. In
> particular, sample probing aims to **fully train** a tentative model and uses its application to
> novel samples as a loss that measures the quality of the samples, computed **at every single
> generative model update step** in an **efficient and end-to-end differentiable** way.
>
> We are unaware of any other approach that leverages closed-form predictive models for sample probing to
> define a task-dependent loss for generative GZSL model training.
>
> We believe that the revised manuscript also much better presents the novelty of the proposed approach as a
> general technique to improve generative GZSL model training. Most notable changes that
> better highlight the contribution:
> * Much more comprehensive evaluation of the approach over multiple generative models (Table 1).
> * Evaluation of the approach's usability with alternative probe models (Table 2), demonstrating the
>   generality of the approach.
> * The following method summary and discussion is added to the end of Section 3: *The proposed
>   approach aims to realize the goal of learning to generate good training samples by evaluating the
>   synthesis quality through the lens of a closed-form trainable probe model, the prediction loss of
>   which is used as a loss for the G updates. Therefore, G is expected to be progressively guided
>   towards producing realistic, relevant and informative samples, through the reinforcement of which
>   may vary depending on the inherent nature of the chosen probe model.*
>
> ### Q2. "The closed-form probe model is borrowed from [a],[b],[c]. ([a]- Meta-Learning for Generalized Zero-Shot Learning, AAAI 2020. [b]- Episode-Based Prototype Generating Network for Zero-Shot Learning, CVPR 2020. [c]- Towards Zero-Shot Learning With Fewer Seen Class Examples, WACV 2021.)"
>
> We believe that the aforementioned improvements and explanations will better highlight the
> distinction of our work. We also note that the closed-form probe model is not a part of [a],[b],[c], and
> overall our work significantly differs from them. The main differences can be summarized as follows:
>
> * [a] (Verma et al., Meta-Learning for Generalized Zero-Shot Learning, AAAI 2020) and [c] (Verma et
>   al., Towards Zero-Shot Learning With Fewer Seen Class Examples, WACV 2021) are pioneering works
>   on the use of meta-learning principles for generative GZSL model training.  However, these works
>   follow a different direction by integrating Model-Agnostic Meta-Learning (MAML) update rules.  At
>   each step, a single-step update is first tentatively applied to the model components over real
>   seen class training examples as the inner-loop update, and the loss of the updated components is
>   then computed over real unseen class samples. The resulting compute graph is then used to make a
>   permanent gradient descent/ascent update step. In contrast, at each step, our proposed approach
>   samples a batch of synthetic examples from the generative model, **fully trains** a tentative ZSL
>   model using the synthetic examples, and computes the tentative ZSL model's loss on novel real examples
>   to be used as a loss for updating the generative model. The complete training of a tentative model
>   per step and end-to-end differentiability of the whole compute chain is made possible by the idea
>   of leveraging closed-form solvable formulations as the tentative sample probe models, which has
>   not been explored in any prior work that we are aware of.
>   Therefore, the foundations of [a,c] versus ours are completely different other than embracing a form of
>   meta-learning principles.
>
> * [b] (Yu et al Episode-Based Prototype Generating Network for Zero-Shot Learning, CVPR 2020): this
>   work differs categorically from ours as it does not aim to train a generative model for sample
>   generation. The prototype generation refers to learning a deterministic mapping from attributes to
>   class prototypes, much like the attribute-to-visual mapping of a bi-linear compatibility function.
>   Therefore, we do not see any close relation to ours.
>
> Finally, we also note that Section 2 provides a detailed discussion of these three works. We will be
> happy to further improve their discussion upon suggestions.

---

> ### Author Response · Authors · 2021-11-23
> **Reply to Reviewer vUDu (continued)**
>
>
>
> ### Q3. "To prove the better efficacy of the proposed model, it should be trained using a few examples (5/10 samples) per seen class."
>
> Learning a generative GZSL model from a few examples per class is an interesting research direction,
> but it is outside the scope of our work. We follow the experimental setting of the mainstream works
> on generative models for GZSL. We also note that learning generative GZSL models with few samples is
> not a commonly studied problem in generative GZSL. Towards Zero-Shot Learning With Fewer Seen Class
> Examples (Verma et al, WACV 2021) is the only work that we are aware of in this direction.
>
> ### Q4. "The experiments for a large dataset (ImageNet) should be included for better evaluation."
>
> Many thanks for the comment. We indeed tried to do ImageNet-scale evaluation, however, so far we
> could not obtain a strong generative GZSL baseline using our computational
> resources. We also tried to bypass this problem by leveraging an existing implementation,
> however, none of the official source codes that we could find for generative GZSL papers with ImageNet results
> (e.g. f-VAEGAN-D2, Feature Generating Networks for Zero-Shot Learning, Multi-modal Cycle-consistent
> Generalized Zero-Shot Learning) provide the configuration to re-produce ImageNet results.
>
> We also note that while ImageNet evaluation is interesting, it has very different challenges, and
> methods well-performing on GZSL benchmark datasets may not perform well on ImageNet, vice versa.
> Therefore, we believe that ImageNet evaluation should not be seen as a requirement for proper
> validation. In fact, the papers of most of the recent and/or state-of-the-art generative GZSL
> models, e.g.  LisGAN (CVPR 2019), TF-VAEGAN (ECCV 2020), META-VGAN (WACV 2021) FREE (ICCV 2021)
> (among others), do not report ImageNet evaluation results.

---

### Author Response · Authors · 2021-11-23
**Summary of changes**

# List of changes

We thank all the reviewers for their detailed evaluation of the manuscript. We have carefully
revised the paper to incorporate all the comments, questions and suggestions. Thanks to the
comments, the paper is now significantly improved in a number of ways, most importantly in terms of
clarifying the contributions and providing a much more comprehensive experimental validation. We
believe that the revised paper (jointly with our replies) addresses all of the main points raised in the
reviews.

### Highlights
Experimental evaluation has been greatly improved according to the review suggestions in three
  main ways:
1) The proposed sample probing approach is now evaluated over
5 different generative models (4 formulations, plus one model over fine-tuned features), using the
4 benchmark datasets. The results are presented in Table 1, where we observe improvements in 17 out
of 19 h-scores using the proposed approach.
2) The experimental validation is further improved by adding two additional closed-form probe models.
Old Table 2 (cWGAN-only result) is replaced by a completely new table to report the corresponding
experiments. The results overall demonstrate the applicability of the proposed approach with
alternative probe models.
3) t-SNE visualizations are added to Appendix B, which qualitatively show how sample probing can improve
the predicted sample distributions of unseen classes.

### Detailed list of changes
* Table 1 now presents evaluation of sample probing (with ESZSL) in combination with multiple
generative GZSL models. While we had previously focused only on the TF-VAEGAN baseline, we now
evaluate the sample probing approach on top of the LisGAN (CVPR 2019) and FREE (ICCV 2021), in
addition to TF-VAEGAN results.  We have also extended the cWGAN evaluation, and updated the
corresponding discussions in Section 4. Overall, we observe that sample probing improves
h-scores in 17 out of 19 cases, which provides a much stronger validation of the approach.

* We had previously used only ESZSL as the closed-form probing model.   We now evaluate the
approach with two additional closed-form probing model alternatives (called Sim2Vis and
Vis2Sim). We added their descriptions to Section 3.3, the experimental results to Table 2
and detailed discussions into Section 4.1. The successful results provide a validation for
the versatility of the approach in terms of compatibility with different closed-form probe
models other than ESZSL.

* In Figure 2, ESZSL solver box is replaced with "Probe model solver" to avoid confusion,
after the inclusion of two additional sample probe models. We have also found and fixed a
typo regarding the dimensions of the matrix A in ESZSL definition in Section 3.3.

* We have added per-class tSNE plots of real and generated samples, with and without
sample probing, to Appendix B. The plots, overall, demonstrate how sample probing can
improve the overall sample quality of a generative model.

* We moved the table with general comparisons to other state-of-the-art generative GZSL
models to appendix, (i) due to the page limit after the aforementioned additions, (ii)
for its lesser importance due to the crudeness of the comparisons. A brief discussion
on this and a reference to the appendix section is added to Section 4.1.

* We have made several minor revisions and language improvements throughout the paper with no
changes in meaning or loss of information, to improve the text and fit into the page
limit.

* An extended summary has been added to the end of Section 3 (Method). Section 5 (Conclusions) has
  been revised to reflect the newly added experimental results.

* Table 3 now presents the baseline (w/o sample probing) for easier evaluation of zsl-loss versus gzsl-loss.

* Finally, we note that while a large number of new results based on a state-of-the-art generative
  models have been added to Table 1, only the AWA result for cWGAN is currently missing, simply
  because we need to tune its all hyper-parameters from scratch and we have not been reached a
  competitive baseline (without sample probing) yet in our limited model selection experiments so
  far. We note that cWGAN is the simplest and chronologically the oldest baseline among all.

---

### Decision · Program_Chairs · 2022-01-20

**Decision:**

Accept (Poster)

**Comment:**

The paper proposes to improve (generalized) zero-shot learning, by training a generator jointly with the classification task, such that it generates samples that reduce the classification loss.  To achieve this, they use a zero shot model that has a (differentiable) closed form solution (ESZSL), so the full model can be optimized end-to-end. The approach is evaluated on the standard benchmarks of GZSL.

Reviewers had some concerns regarding novelty compared with previous work and quality of experiments and evaluations. The authors answered most of these concerns in their rebuttal including discussion with previous work and additional evaluations.  As a result, the paper would be interesting for the ICLR audience.